# RNA Viruses of *Amblyomma variegatum* and *Rhipicephalus microplus* and Cattle Susceptibility in the French Antilles

**DOI:** 10.3390/v12020144

**Published:** 2020-01-26

**Authors:** Mathilde Gondard, Sarah Temmam, Elodie Devillers, Valérie Pinarello, Thomas Bigot, Delphine Chrétien, Rosalie Aprelon, Muriel Vayssier-Taussat, Emmanuel Albina, Marc Eloit, Sara Moutailler

**Affiliations:** 1UMR BIPAR, Animal Health Laboratory, ANSES, INRAE, Ecole Nationale Vétérinaire d’Alfort, Université Paris-Est, 94700 Maisons-Alfort, France; mathilde.gondard@gmail.com (M.G.); elodie.devillers@anses.fr (E.D.);; 2CIRAD, UMR ASTRE, F-97170 Petit-Bourg, Guadeloupe, France; valerie.pinarello@cirad.fr (V.P.); rosalie.aprelon@cirad.fr (R.A.); emmanuel.albina@cirad.fr (E.A.); 3Pathogen Discovery Laboratory, Inserm U1117, Biology of Infection Unit, Institut Pasteur, 75015 Paris, France; sarah.temmam@pasteur.fr (S.T.); thomas.bigot@pasteur.fr (T.B.); Delphine.chretien@pasteur.fr (D.C.); 4ASTRE, University Montpellier, CIRAD, INRAE, 34000 Montpellier, France; 5Bioinformatics and Biostatistics Hub, Computational Biology Department, Institut Pasteur, USR 3756 CNRS, 75015 Paris, France; 6National Veterinary School of Alfort, Paris-Est University, Maisons-Alfort, 94704 Cedex, France

**Keywords:** ticks, cattle, RNA viruses, next-generation sequencing, phylogeny, microfluidic real-time PCR technology, Caribbean, LIPS

## Abstract

Ticks transmit a wide variety of pathogens including bacteria, parasites and viruses. Over the last decade, numerous novel viruses have been described in arthropods, including ticks, and their characterization has provided new insights into RNA virus diversity and evolution. However, little is known about their ability to infect vertebrates. As very few studies have described the diversity of viruses present in ticks from the Caribbean, we implemented an RNA-sequencing approach on *Amblyomma variegatum* and *Rhipicephalus microplus* ticks collected from cattle in Guadeloupe and Martinique. Among the viral communities infecting Caribbean ticks, we selected four viruses belonging to the *Chuviridae*, *Phenuiviridae* and *Flaviviridae* families for further characterization and designing antibody screening tests. While viral prevalence in individual tick samples revealed high infection rates, suggesting a high level of exposure of Caribbean cattle to these viruses, no seropositive animals were detected. These results suggest that the *Chuviridae*- and *Phenuiviridae*-related viruses identified in the present study are more likely tick endosymbionts, raising the question of the epidemiological significance of their occurrence in ticks, especially regarding their possible impact on tick biology and vector capacity. The characterization of these viruses might open the door to new ways of preventing and controlling tick-borne diseases.

## 1. Introduction

Ticks harbor a wide variety of microorganisms, such as nematodes, fungi, protozoa, bacteria, and viruses [1,2]. To date, about 160 arboviruses have been identified in ticks, with around 25% of them associated with human and/or animal diseases [3]. Arboviruses are usually grouped into nine viral families: one family of DNA viruses, *Asfarviridae*, and eight families of RNA viruses, *Flaviviridae*, *Orthomyxoviridae*, *Reoviridae*, *Rhabdoviridae*, *Nyamiviridae*, *Nairoviridae*, *Phenuiviridae*, and *Peribunyaviridae* [3].

Studies on tick-borne viruses have mainly focused on arboviruses that are able to affect both the invertebrate and vertebrate host, and that are responsible for important human or animal diseases worldwide [3,4]. However, over the last decade, with advances in next-generation sequencing and the growing interest in arthropod microbiome characterization, several studies have revealed how far we are from fully understanding the virome diversity in arthropods [5,6,7,8,9]. New viruses have been described worldwide in various arthropods, revealing highly variable genomic structures and genetic organization, defining new viral families, and revealing complex evolutionary links with the known arbovirus families and genera [7,10,11,12].

Regarding ticks, virome high-throughput sequencing has been performed in various species from Asia [7,13,14,15], North America [16,17,18], South America [19,20], Europe [8,9], Africa [21], Australia [22] and Trinidad and Tobago [23], all revealing extensive diversity in RNA viruses [24]. Characterizing these new viruses offers new perspectives to better understand viral origins and evolution, for which arthropods seem to play a key role, for example by allowing interactions and genetic exchanges within their “virosphere” [7,14]. These findings also raise important questions regarding the impact of these viruses in human or animal health. In fact, the ability of such viruses to infect vertebrates and their pathogenicity remain to be elucidated in most cases. Interestingly, as potential tick endosymbiotic components, the arthropod virome might play a role in vector biology or pathogen transmission, as has been described for endosymbiotic bacteria [24,25,26]. Therefore, deciphering the complex interactions between the arthropod and its virome seems to be a promising challenge that might radically transform control strategies for arthropod-borne pathogens and vectors [26,27].

In the Caribbean, despite the importance of tick-borne diseases in animal health management, very little research has been carried out on tick-borne viruses [23,28]. Cases of African swine fever (ASFV, *Asfarviridae*) were described in Cuba, Haiti, and the Dominican Republic in the 1970s, and viruses have been described in rare reports concerning ticks parasitizing seabirds, such as Estero real, Hughes and Soldado nairoviruses (*Nairoviridae*) [29,30,31,32,33,34]. Recently, virome analyses of ticks collected in Trinidad and Tobago provided the first update on viral communities infecting Caribbean ticks, with the report of nine viruses belonging to the *Tymovirales*, *Bunyavirales, Chuviridae*, *Rhabdoviridae* and *Flaviviridae* [23].

Here, we report on the virome analysis of the two main tick species involved in tick-borne diseases in the Lesser Antilles, *Amblyomma variegatum* and *Rhipicephalus microplus.* Ticks were collected from cattle in the French Antilles, and the virome was analyzed using a metatranscriptomic approach. The prevalence of the four most abundant viruses (including members of the *Chuviridae*, *Phenuiviridae*, and *Flaviviridae*) was determined at the individual tick level using high-throughput microfluidic real-time PCR technology, and evaluation of cattle exposure to these viruses was determined by antibody screening using a Luciferase Immunoprecipitation System (LIPS) assay.

## 2. Materials and Methods 

### 2.1. Ticks and Cattle Sera Collected in Guadeloupe and Martinique

A total of 578 adult ticks were collected on cattle in Guadeloupe and Martinique. In Guadeloupe, 132 adult *Amblyomma variegatum* and 165 adult *Rhipicephalus microplus* specimens were sampled between February 2014 and January 2015, from 40 cattle originating from 22 different herds. In Martinique, 281 adult *Rhipicephalus microplus* specimens were collected between February and March 2015, from 29 cattle originating from 29 herds. All the ticks were collected from cattle, partially engorged, and then stored at −80 °C [35]. Ticks were morphologically identified to the species level [36]. Finally, 178 and 22 cattle sera were collected in Guadeloupe in 1994–1995 and in 2019, respectively. Sera were obtained from disease surveillance campaigns undertaken by the local representative of the French Ministry in charge of Agriculture and approved by the animal owners. Ticks were collected on cattle with the agreement and the help of animal owners in 2015 (PathoID project “Rodent and tick Pathobiome”, between 2014–2015, grant number PATHO-ID Metaprogram MEM 2012–2014 and Resist project “Assessment of Tick Resistance to Acaricides in the Caribbean”, in 2015, grant number FCR2013/02).

### 2.2. Nucleic Acid Extraction

For nucleic acid extraction, 1 mL of freshly prepared PBS 1X was added to 20 mg of ticks. Concerning *Amblyomma variegatum* ticks collected in Guadeloupe, one sample corresponded to one adult specimen leading to the constitution of 132 tick samples. Regarding *Rhipicephalus microplus* from Guadeloupe and Martinique, as some adult specimens were under the required weight threshold, pools of one to four adult ticks were constituted, leading to the formation of 391 tick samples. Thus, a total of 523 adult tick samples were processed for this study. Samples were then shaken for 3 min at 7 Hz/s with a TissueLyzer (Qiagen, Hilden, Germany) as a washing step. Supernatants were discarded and then the ticks were frozen at −80 °C for 20 min. After the addition of a steel bead, the samples were crushed twice for 2 min at 30 Hz/s with a TissueLyzer (Qiagen, Germany). 450 µL of fresh PBS 1X were added to the samples, which were then vortexed for 10 s, and centrifuged for 3 min at 8000× *g*. Finally, 20 µL of Proteinase K were added to 150 µL of crushed tick sample, and RNA was extracted using a NucleoSpin^®^ 96 Virus Core Kit (Macherey-Nagel, Düren, Germany) and the automatic platform Biomek4000 (Beckman Coulter, Villepinte, France). DNA and RNA were simultaneously extracted according to the manufacturer instructions with the exception of the addition of poly-A RNA carrier in the lysis buffer which was replaced by 4 µL of linear polyacrylamide with an initial concentration of 5 mg/mL (Ambion, Paris, France). Total nucleic acid per sample was eluted in 160 µL of elution buffer and stored at −80 °C until further use.

### 2.3. High-Throughput Sequencing, Bioinformatic and Phylogenetic Analyses

For high-throughput sequencing, 5 µL of each nucleic acid sample were pooled, and DNA was digested using a TURBO DNA-free™ Kit (Invitrogen, Carlsbad, CA, USA), according to the manufacturer’s instructions. Purified RNA was used as a template for reverse transcription using random hexamers, followed by random amplification using a Qiagen QuantiTect Whole Transcriptome Kit. cDNA was used for library preparation using a TruSeq stranded total RNA Kit with RiboZero. The library was sequenced onto an Illumina NextSeq 500 sequencer in a paired-ends 2 × 75 bp format, outsourced to DNAVision (Charleroi, Belgium).

Raw reads were processed with an in-house bioinformatics pipeline, as previously described [15]. Briefly, raw reads were trimmed, assembled into contigs which were together with remaining singletons translated into proteins (https://figshare.com/articles/translateReads_py/7588592). Taxonomic assignments of sequences was performed by an initial BlastP similarity search against the protein Reference Viral database (RVDB [37]), followed for all viral hit by a BlastP against the whole protein NCBI/nr database (release 10/25/2018). The quantification of abundance of each viral taxon was obtained by summing the length (in amino acids) of all sequences associated with this taxon, and weighted by the *k*-mer coverage of all contigs associated with a given taxon [15].

Phylogenetic reconstructions of *Bunyavirales* and *Mononegavirales*/*Jingchuvirales* evolution were performed on the conserved non-structural RNA-dependent RNA polymerase gene (RdRP), as previously described [15]. Briefly, complete open reading frames (ORFs) were aligned with Multiple Alignment using Fast Fourier Transform (MAFFT) aligner under the L-INS-I parameter [38]. Alignments were cured with the BMGE 1.12 tool, implemented through the NGPhylogeny portal [39,40]. The best amino acid substitution models that fitted the data were determined with ATGC Start Model Selection [41], implemented in http://www.atgc-montpellier.fr/phyml-sms/ using the corrected Akaike information criterion. Phylogenetic trees were constructed using the maximum likelihood (ML) method, implemented through the RAxML program under the CIPRES Science Gateway portal [42] according to the selected substitution model. Nodal support was evaluated using the “automatic bootstrap replicates” parameter.

The complete ORFs were obtained using conventional PCR and Sanger sequencing after designing specific primers targeting the identified viruses, as previously described [15]. Briefly, viral RNA was reverse transcribed using SuperScript IV reverse transcriptase (Invitrogen, USA), and cDNA was subsequently used to fill the gaps in the genomes using Phusion High Fidelity DNA polymerase (New England Biolabs, Évry, France). Positive PCR products were further purified and sequenced by Sanger sequencing on the Eurofins Segenic Cochin platform. When start and stop codons were lacking, RACE-PCR was performed using the 5′/3′ RACE kit, 2nd Generation (Roche Applied Science, Penzberg, Germany).

### 2.4. Tick-Borne Virus Screening in Ticks from the French Antilles

Viruses of medical and veterinary importance, as well as the four new viruses described by sequencing, were monitored in individual RNA tick samples. RNA samples were retro-transcribed to cDNA, using a qScript cDNA Supermix Kit (Quanta Biosciences, Beverly, MA, USA), and then PerfeCTa^®^ PreAmp SuperMix (Quanta Biosciences, Beverly, MA, USA) was used for cDNA pre-amplification, following the manufacturer’s instructions. High-throughput microfluidic real-time PCR amplifications were performed using a BioMark™ real-time PCR system (Fluidigm, South San Francisco, CA, USA), and 48.48 dynamic arrays (Fluidigm, CA, USA), allowing the detection of 22 viral pathogens as described in Gondard et al., [43], with the addition of PCR systems specifically targeting the four new viruses described by sequencing (Table 1, and Appendix A). For each detected pathogen, infection rates were estimated according to the tick species and the island of origin. Infection rates were defined as the proportion of infected ticks over the total number of ticks analyzed. Most of the samples were single specimens of ticks, but since 49 samples consisted of a pool of 2 to 4 tick specimens, infection rates were expressed as the minimum and maximum proportions of infected ticks, out of the total number of ticks analyzed.

### 2.5. Endogenous Viral Element Analysis

Karukera tick virus (KTV), Wuhan tick virus 2 (WhTV2), Lihan tick virus (LTV), and Jingmen tick virus (JMTV) were screened in the DNA of individual tick samples in order to identify possible endogenous viral elements (EVEs). Tick nucleic acids were not RNA retro-transcribed but directly processed with PerfeCTa^®^ PreAmp SuperMix (Quanta Biosciences, Beverly, USA) for DNA pre-amplification, following the manufacturer’s instructions. High-throughput microfluidic real-time PCR amplifications were performed using a BioMark™ real-time PCR system (Fluidigm, USA) and 96.96 dynamic arrays (Fluidigm, USA), as described in Gondard et al., [43], with the addition of primers and probes specifically targeting the four viruses described by sequencing (Table 1). For each detected pathogen, the infection rates were estimated as previously described for tick-borne virus screening in ticks from the French Antilles.

The comparison of the mean Cp values obtained for each virus when testing RNA and DNA samples was performed with Student’s t-test and R version 3.6.0 (26 April 2019).

### 2.6. Serological Screening of Cattle Exposed to Tick Bites

The identification of putative viral antigenic regions was performed as previously described [15]. To maximize the probability of detecting cattle antibodies specific to Karukera tick virus (KTV), Wuhan tick virus 2 (WhTV2), or Lihan tick virus (LTV), we targeted either extracellular domains of the glycoprotein (GP) of KTV and WhTV2, or the nucleoprotein (NP) of LTV—as this virus lacks an M segment, which usually codes for viral GPs. These domains were cloned into a pFC32K vector (Promega, Charbonnières-les-Bains, France) using a Gibson Assembly Kit (NEBuilder^®^ HiFi DNA Assembly Master Mix, New England Biolabs, Evry, France), according to the manufacturer’s instructions. Positive clones were screened by PCR with primers designed in the vector and flanking the inserts, and verified by Sanger sequencing. A PureLink HiPure Midiprep Kit (Invitrogen, USA) was used to extract plasmids from 100 mL of bacterial cultures grown overnight.

HEK-293A cells (kindly provided by Bernard Klonjkowski, Alfort Veterinary School, Maisons-Alfort, France) were transfected with Polyethylenimine (PEI, Polyscience Inc., Tebu-Bio S.A., Le Parray-en-Yvelines, France), as previously described [15]. Briefly, 4 × 10^5^ cells were transfected with 5 µg of plasmid DNA and 20 µL of 1 mg/mL PEI in DMEM medium, supplemented with 1% sodium pyruvate and 1% non-essential amino acids (Invitrogen, USA). Two days post-transfection, fusion proteins were harvested as crude cell lysates [15]. Luciferase activity was measured on a Centro XS^3^ LB 960 Luminometer (Berthold Technologies, Thoiry, France).

A LIPS assay was performed as described by Burbelo et al. [44], except that cattle sera were not diluted. The residual background was measured as the mean of 8 negative controls (without serum), and the positivity threshold was defined as the mean of these controls + 5 standard deviations.

## 3. Results

### 3.1. Virome Composition of Caribbean Cattle-Associated Ticks

The RNAseq analysis of the pools of RNA extracted from 578 ticks, including 132 *Amblyomma variegatum* and 446 *Rhipicephalus microplus* collected in Guadeloupe and Martinique, provided 41,696,475 paired-end reads, generating 28,565 contigs and 1,188,734 singletons, after trimming and assembly. Of these, most viral sequences were assigned to ssRNA viruses (99.8%), while dsRNA viruses were in the minority (0.2%), and included only *Partitiviridae*-related sequences. No transcripts associated with a DNA virus were identified. Positive sense RNA viruses were the most abundant (95%) and comprised viral genomes belonging to the *Flaviviridae*, *Solemoviridae*, and *Tymoviridae* families, while negative sense RNA viruses (5% of ssRNA viruses) were assigned to the *Phenuiviridae* and *Chuviridae* families (Table 2). Among the *Flaviviridae*, the only viral genome detected was related to Mogiana tick virus, a tick-associated Jingmen virus primarily described in *Rhipicephalus microplus* ticks from Brazil [19,45]). As the genome of the Jingmen tick virus found in ticks from Guadeloupe and Martinique has already been characterized in a previous study, and serological screening in cattle blood has been performed [46], only the infection rates are described in this paper (see Section 3.2).

#### 3.1.1. Viruses Belonging to the Chuviridae Family

Nearly six-thousand (5653) reads were assigned to the Changping tick mivirus and assembled into a virus genome tentatively named Karukera tick virus (KTV, accession number MN599998). KTV, a new mivirus member of the *Chuviridae* related to Changping tick virus 2 and Brown dog tick mivirus 1, displayed an unsegmented circular RNA genome of 11,177 nucleotides with three ORFs that encode the RNA-dependent RNA polymerase (RdRP), the glycoprotein (GP), and the nucleoprotein (NP) of the virus. The RdRP gene comprises two functional domains (one related to the *Mononegavirales* RdRP and the second to the *Paramyxoviridae* mRNA capping enzyme), while the GP comprises one functional domain (Figure 1a). Karukera tick virus displayed quite low levels of sequence identity with its closest relative Brown dog tick mivirus 1 (Table 3), sharing only 82%, 77% and 62% amino acid identities with Brown dog tick mivirus 1 in the RdRP, GP and NP genes, respectively. Brown dog tick mivirus 1 was previously identified in *Rhipicephalus sanguineus* ticks from Trinidad and Tobago [23]. In addition, KTV displayed lower identity with the Changping tick virus 2 prototype strain, previously reported in *Dermacentor* spp. ticks from China (for example, 63.96% sequence identity in the RNA polymerase with Changping tick virus 2 (YP_009177704.1). This low level of sequence identity suggests the presence of a new *Chuviridae* member in the Caribbean ticks. Phylogenetic analyses performed on the complete RdRP protein confirmed that Karukera tick virus belongs to circular *Chuviridae* viruses, and in particular to the Changping mivirus species (Figure 2).

More than sixty-thousand (63,391) reads were assigned to Wuhan mivirus. Wuhan tick virus 2 (WhTV2, accession number MN599999) identified in French Antilles ticks presented, as expected, an unsegmented circular RNA genome of 11,393 nucleotides, also displaying the three typical ORFs of the *Chuviridae* (Figure 1b), respectively coding for the viral RdRP (2189 aa), the viral GP (683 aa), and the viral NP (411 aa). Like its closest relative, WhTV2 displayed two functional domains in the RdRP (related to *Mononegavirales* polymerase and mRNA capping enzymes) and one domain in the GP (related to pseudorabies glycoprotein B). The Caribbean strain of WhTV2 shared 99.4% nucleotide identity with Wuhan tick virus 2 described in *Rhipicephalus microplus* ticks from Brazil (Table 3). This high level of sequence identity suggests the presence of a new genotype of WhTV2 in Caribbean ticks. Phylogenetic analyses performed on the complete RdRP protein placed the Caribbean WhTV2 strain in the Wuhan mivirus species (Figure 2). Of note, all WhTV2 genomes originated from *Rhipicephalus microplus* ticks, suggesting a possible restriction to this tick species. Interestingly, however, with a high supported node of 98, all WhTV2 variants originating from Brazil, Trinidad and Tobago, and the French Antilles clustered together in a sub-clade distinct from Asian strains (China and Thailand), suggesting distinct evolution related to the geographic origin (Figure 2).

#### 3.1.2. Viruses Belonging to the *Phenuiviridae* Family

More than twenty-thousand (23,345) reads were assigned to Lihan tick virus (LTV, accession numbers MN599996 and MN599997), previously reported in *Rhipicephalus microplus* ticks from Brazil (Figure 1c, Table 3). The LTV strain identified from French Antilles ticks presented two segments, displaying the typical ORF of the *Phenuiviridae* family, including the ORF coding for the viral RNA-dependent polymerase located on the L segment, and the ORF coding for the nucleoprotein located on the S segment. LTV viral proteins displayed the characteristic functional domains of bunyaviruses, i.e., the endonuclease and polymerase activities carried by the RdRP protein, and the *Phlebovirus*/*Tenuivirus* nucleocapsid protein carried by the S segment (Figure 1c). The Caribbean LTV variant displayed high levels of nucleotide identity (98%–99% depending on the segment) with LTV variants either originating from *Rhipicephalus microplus* ticks from Brazil (L segment) or China (S segment) (Table 3), suggesting the identification of a new genotype of LTV in Caribbean ticks. This also suggests a high conservation level between geographically distant isolates, as confirmed by phylogenetic analyses which did not present clear clustering of different LTV strains according to their geographic origins (Figure 3). Interestingly, LTV strains clustered together in a distinct clade (restricted to tick-borne viral genomes coming from the USA, China, Turkey, Thailand, Brazil, and Trinidad and Tobago) that was positioned at the root of known tick-borne and mosquito/sandfly-borne phleboviruses, suggesting a possible tick origin of known phleboviruses, as previously suggested [7] (Figure 3, inset).

### 3.2. Screening of Tick-Borne Viruses in Individual Tick Samples from Guadeloupe and Martinique

In all, 132 *Amblyomma variegatum* collected in Guadeloupe, and 446 *Rhipicephalus microplus*, including 165 from Guadeloupe and 281 from Martinique were tested with a microfluidic real-time PCR system for the screening of tick-borne viruses of medical and veterinary importance, and to monitor Karukera tick virus (KTV), Wuhan tick virus 2 (WhTV2), Lihan tick virus (LTV), and Jingmen Tick virus (JMTV). Among the ticks’ samples analyzed here, none of the 22 viruses of medical or veterinary interest belonging to the viral families *Asfarviridae*, *Orthomyxoviridae*, *Reoviridae*, *Bunyaviridae* and *Flaviviridae*, were detected (see list of the targeted virus in Appendix A). However, Karukera tick virus, Wuhan tick virus 2, Lihan tick virus, and Jingmen Tick virus identified by NGS were found to be widely distributed among the tick samples of Guadeloupe and Martinique (Figure 4).

KTV was only found in Guadeloupe, in 23% of *Amblyomma variegatum* ticks and in only one sample of *Rhipicephalus microplus* (Figure 4). WhTV2 was detected in 12.6% of *Amblyomma variegatum* ticks, and in at least 63.6% and 92.5% of *Rhipicephalus microplus* collected in Guadeloupe and Martinique, respectively (Figure 4). The L and S segments of LTV were mostly detected simultaneously across the positive samples. In Guadeloupe, 11.9% and 10.4% of *Amblyomma variegatum* ticks and at least 63.6% and 64.8% of *Rhipicephalus microplus* ticks were found to be positive for both the L and S segments, respectively. In Martinique, up to 83.6% and 90.7% of *Rhipicephalus microplus* were positive for both the L and S segments, respectively (Figure 4). Finally, JMTV was found in both tick species originating from the two islands. However, the infection rates obtained were not consistent depending on the targeted segment. Infection rates obtained when targeting segments 1 or 3—which encode the nonstructural proteins—were clearly lower than those obtained when targeting segments 2 or 4, which encode the structural proteins (Figure 4). This result suggests either a difference in the sensitivity of the PCR and/or a difference in terms of quantity and expression between the various RNA segments, as observed for example in bunavirales RNA Segments [47]. This could explain why in this study, segment 1 of JMTV—which encodes the nonstructural protein that corresponds to the RNA polymerase—is under-expressed and thus detected to a lower degree than segments 2 and 4, which encode the structural proteins. Regarding the results obtained when targeting segment 4, JMTV was detected in 5.2% of the *Amblyomma variegatum* and in at least 24.2% and 76.9% of the *Rhipicephalus microplus* from Guadeloupe and Martinique, respectively (Figure 4).

### 3.3. Viral Co-Infections

The majority of positive *Amblyomma variegatum* samples demonstrated single infection, with 41% with KTV, 18% with LTV, and 11% with WhTV2 (Figure 5). Conversely, most positive *Rhipicephalus microplus* ticks presented coinfections, with the detection of two to four viruses within the same tick sample (Figure 5). In Guadeloupe, 49% and 37% of *Rhipicephalus microplus* were infected by the combination of WhTV2/LTV and WhTV2/JMTV/LTV, respectively. In Martinique, WhTV2/JMTV/LTV triple infection represented up to 76% of positive *Rhipicephalus microplus* samples, and WhTV2/LTV co-infection 14% (Figure 5). The two tick species, *Amblyomma variegatum* and *Rhipicephalus microplus*, seemed to display two different viral infection/co-infection patterns. *Amblyomma variegatum* samples were found mostly infected with KTV, and most of the infections were single infections. Inversely, WhTV2, LTV and JMTV were mainly found in *Rhipicephalus microplus* ticks and in most cases, double or triple coinfections were the rule. These results suggested that the level and nature of coinfections could be influenced by the tick species.

### 3.4. Search for Endogenous Viral Elements

Endogenous viral elements (EVEs) correspond to the integration of viral DNA fragments (or cDNA fragments in the case of RNA viruses) in the genome of the host, and have been described in many eukaryotic genomes, including arthropods and ticks [48,49]. Although endogenous viral sequences are generally described as non-functional pseudogenes or fossil DNA, some EVEs encode intact ORFs that can be expressed [50]. This may be related either to recent endogenization of the viral genome in the host genome, or to exaptation (positive selection) of the EVE, during the evolution of the host genome [50]. Thus, even if all the viral ORFs found in this study were complete, we could not eliminate the possibility of EVE in our samples. In order to explore the potential presence of integrated viral sequences into tick genomes that could have been sequenced, we screened the 523 corresponding tick DNA samples (Table 4). All *Amblyomma variegatum* ticks were found to be negative for the presence of KTV-, WhTV2-, LTV-, and JMTV-related viral sequences, while some *Rhipicephalus microplus* ticks from both Guadeloupe and Martinique were found to be positive (Table 4). The presence of KTV-related DNA was only detected in one *Rhipicephalus microplus* tick from Guadeloupe, while WhTV2-related DNA was identified in up to 35.8% and 32% of *Rhipicephalus microplus* ticks from Guadeloupe and Martinique, respectively. Similarly, LTV-related DNA was detected in up to 1.2% and 5.7% of *Rhipicephalus microplus* ticks from Guadeloupe and Martinique, respectively (data based on Segment L detection, Table 4). Finally, JMTV-related DNA fragments were found in up to 1.2% and 5% of *Rhipicephalus microplus* ticks from Guadeloupe and Martinique, respectively (data based on Segment 4 detection, Table 4). Overall, the detection of KTV-, WhTV2-, LTV-, and JMTV-related viral sequences in tick DNA samples showed lower infection rates compared to rates obtained in tick RNA samples (Appendix A and Table 4). In addition, the Cp values of targeted DNA viral sequences were significantly much higher than those obtained for targeted RNA sequences (Appendix A). Together, these observations suggest either that the detection of the viruses in DNA samples may be the result of residual RNA contamination—as both nucleic acids were extracted simultaneously—or the potential co-occurrence, at a low level, of endogenous viral elements in tick genomes along with exogenous viral particles.

### 3.5. Serological Screening of Guadeloupean Cattle Exposed to Tick Bites

To test whether KTV, WhTV2 and LTV were able to infect cattle highly exposed to tick bites, and therefore could constitute putative novel tick-borne arboviruses, we developed LIPS-based serological screening against these viruses. None of the cattle sera presented luciferase activity higher than the positivity threshold, showing that no antibodies against Karukera virus, Wuhan tick virus 2, and Lihan tick virus were detected in cattle sera (Appendix A). The maximum prevalence of sera reacting to at least one of the three viral constructs was estimated to be 0.047% (*p* = 0.05) in cattle, meaning that these infections, if they occur, are very rare (Appendix A).

## 4. Discussion

We performed an analysis of the meta-transcriptome of cattle-infesting ticks from Guadeloupe and Martinique. Despite a high proportion of unassigned sequences, this analysis allowed us to generate an overview of the viruses present in *Amblyomma variegatum* and *Rhipicephalus microplus* ticks from the French West Indies that matched with sequences available in public NCBI databases. In addition to viruses infecting plants or restricted to arthropods, the sequencing data revealed the presence of four viruses either belonging to families known to comprise arboviruses (*Flaviviridae* and *Peribunyaviridae*-related viruses), or viruses for which the ability to infect vertebrates is still unknown (*Chuviridae*-related viruses). These *Chuviridae*-related viruses belong to new arthropod-associated viral groups described since 2014 [7,19,51]. Sequencing results were confirmed by the screening analysis of the individual tick samples by high-throughput microfluidic real-time PCR, including as targets 22 known tick-borne viruses (TBVs) and these four viruses. Only the four viruses identified by NGS, KTV, WhTV2, LTV and JMTV, were detected in individual ticks. The absence of TBVs of medical or veterinary importance in our samples was not surprising given the absence of reports of viral diseases associated with ticks for several decades, except for the particular cases of African swine fever reported in the 1970s in the Caribbean [28,52].

Interestingly, the individual screening of tick samples allowed us to identify different patterns of viral infection according to the tick species, here *Amblyomma variegatum* and *Rhipicephalus microplus*.

*Amblyomma variegatum* ticks collected in Guadeloupe were mainly infected by a new member of the *Chuviridae*, tentatively named Karukera tick virus (23%). The monophyletic *Chuviridae* viral family includes negative-sense RNA viruses presenting various genome organizations, from linear to the circular forms that can be unsegmented or bi-segmented, and phylogenetically located at an intermediate position between segmented and unsegmented RNA viruses [7,20]. KTV clustered with other Changping miviruses that have been identified in different hard tick species worldwide, such as the virus strains from Trinidad and Tobago, and Thailand in *Rhipicephalus sanguineus* ticks [15,22], Chinese strains in *Dermacentor* spp. or *Haemaphysalis parva* ticks [7], or Turkish strains in *Haemaphysalis parva* [53]. The relatively low level of genome sequence conservation between Changping mivirus isolates originating from different tick species reinforces the hypothesis of virus specialization and co-evolution with their respective tick hosts. It has been suggested that a low degree of host restriction, in addition to a high level of virus prevalence in ticks, converge towards the classification of viruses as tick-endosymbionts [15]. However, Sameroff and colleagues suggested that viruses presenting a low degree of host restriction and a high prevalence in tick populations could conversely be considered vertebrate-borne viruses, reflecting the origin of blood meals of ticks [22]. However, the negative serological results for each virus analyzed here, of cattle frequently exposed to tick bites, reinforce the idea of viruses as tick endosymbionts.

*Rhipicephalus microplus* ticks collected in Guadeloupe and Martinique were found to be highly infected with new variants of viruses described in arthropods worldwide, called Wuhan tick virus 2 (WhTV2), Lihan tick virus (LTV) and Jingmen tick virus (JMTV), with infection rates overall higher than 60%. WhTV2 also belongs to the *Chuviridae* family and has, to date, been detected only in *Rhipicephalus microplus* tick species from Brazil, Trinidad and Tobago, China and Thailand [7,15,20,22]. This observation, in addition to high levels of sequence identity between the different variants of WhTV2, suggests a high level of conservation of the virus between *Rhipicephalus microplus* specimens collected worldwide. Interestingly, we observed the formation of two sub-clades according to the geographic origin of the samples, one including the Caribbean and Brazilian variants, the other including the Chinese and Thai variants. An analysis of the co-evolution between WhTV2 and its vector may bring interesting insights on the evolution and dispersion of the *Rhipicephalus microplus* complex worldwide [54,55]. In addition, WhTV2 was the only virus found with an unexpectedly high prevalence in DNA samples from *Rhipicephalus microplus*, suggesting that WhTV2 might be, in addition to exogenous viral particles, an endogenous viral element integrated into the tick genome, with the ability to be transcribed. However, as both DNA and RNA were extracted simultaneously, RNA contamination cannot be ruled out. Lihan tick virus (LTV) belongs to the *Phlebovirus*-like group, with members described in ticks from the USA, Brazil, Trinidad and Tobago, China, and Thailand forming a monophyletic cluster basal to the *Phlebovirus* genus. Viruses belonging to this cluster are characterized by the lack of the M segment, and represent a potential new genus within the *Phenuiviridae* family [7,15,16,18,20,22]. The variant of LTV described here is closely related to the variants found in *Rhipicephalus microplus* from China and Brazil, again suggesting possible vector specificity [7,20]. However, although primarily described in *Rhipicephalus microplus* specimens, LTV has also been reported in Turkish *Hyalomma marginatum* and *Rhipicephalus sanguineus* ticks [53,56], and in Colombian *Dermacentor nitens* ticks (GenBank MK040531). In addition, we also detected LTV in *Amblyomma variegatum* specimens (10.4% to 11.9%). Therefore, the high level of conservation of the LTV variants distributed worldwide and in various tick species suggests that this virus might present a low degree of host restriction.

Several samples of *Amblyomma variegatum* ticks from Guadeloupe were also positive for WhTV2, LTV and JMTV. Likewise, KTV was found in one *Rhipicephalus microplus* sample. Unfortunately, ticks analyzed here were collected partially engorged and most of the *Amblyomma variegatum* and *Rhipicephalus microplus* from Guadeloupe were collected on the same animal. Thus, besides potential low host restriction of the viruses, contamination of the bovine blood meal remaining in engorged ticks or cross-contaminations during co-feeding of the two tick species on the same animal might explain these results. Thus, viral detections described here do not necessarily imply a virus-host relationship. Interestingly, LIPS results did not show any evidence of viral circulation in the vertebrate host, at least for KTV, WhTV2, JMTV and LTV, as none of the bovine sera tested here were found to be positive. The fact that most Guadeloupian cattle sera were collected 20 years before tick sampling may explain this negative serological result, if the introduction of KTV, WhTV2, JMTV and LTV in the region is recent. Molecular clock determination and calculation of the date of introduction may help answer this question. However, similar results were described in the Brazilian study when testing cattle blood samples by RT-PCR targeting the WhTV2 and LTV genomes [20]. Nevertheless, co-feeding transmission of viruses between ticks does not require host viremia, suggesting potential local circulation of the viruses in the vertebrate host. This circulation may be limited temporally and spatially, confined to the tick engorgement site on the host, which would result in virus exchange between ticks without triggering host viremia [57,58,59,60].

Finally, *Amblyomma variegatum* and *Rhipicephalus microplus* also displayed differences in co-infection patterns. Whereas the majority of the positive *Amblyomma variegatum* samples were mono-infected by KTV (41%), the positive *Rhipicephalus microplus* samples displayed high levels of co-infections, with 37% to 76% of the samples triple infected with WhTV2, LTV, and JMTV in Guadeloupe and Martinique, respectively. *Amblyomma* spp. seem to harbor lower viral diversity compared to other tick species, including *Rhipicephalus*, *Ixodes* and *Dermacentor* spp. [17,23]. Bio-ecological parameters may influence virome diversity, such as the tick life cycle or host range [24,61,62,63,64]. For example, although both tick species are mainly found feeding on ruminants in tropical areas, *Amblyomma variegatum* and *Rhipicephalus microplus* present different life cycle strategies. While *Rhipicephalus microplus* has a one-host life cycle, generally spending its whole life feeding on the same host (mainly ruminants), *Amblyomma variegatum* presents a three-host life cycle, meaning that the tick will generally switch from one host to another three times in its life, and early stages (larvae and nymphs) can also feed on small mammals. However, as most of the studies on *Amblyomma variegatum* or *Rhipicephalus microplus* involved ticks collected on cattle, very little is known regarding the diversity of tick hosts in the Caribbean, and this deserves further investigation [28]. Additionally, the composition of the whole microbiome, including bacteria, could influence the differences in viral diversity between the two tick species [25,65,66,67]. Interestingly, *Amblyomma variegatum* specimens collected in Guadeloupe displayed high infection rates for *Rickettsia africae* [35]. It would be interesting to see whether the presence of bacterial endosymbionts, such as *Rickettsia africae* in *Amblyomma variegatum,* could affect the virome diversity of this tick species [68,69].

To conclude, in addition to the characterization of viral genomes identified by NGS, viral isolation of these new viruses should be the next step toward their characterization to determine whether these viruses are exogenous, forming virions in their own right, or whether there are endogenous viral forms, integrated into the genome of the host—in this case the tick [20,48,49,50]. For the moment, only the JMTV has been successfully isolated, and overcoming this crucial step will certainly bring new insights into the biological properties of this virus [51,70]. The interactions between viruses and their respective invertebrate hosts should also be explored, as these viruses (as part of the tick microbiome) could have an impact on tick biology and its ability to transmit pathogens [24,25,26]. Finally, the evaluation of their potential pathogenicity—ability to be transmitted, virulence factors, etc.—for vertebrate hosts should be studied, as some of them, such as the JMTV, have already been involved in human diseases [12,70,71,72]. Deciphering the complex mechanisms governing host–microbiome interactions could eventually help to find new and innovative ways to prevent and control tick-borne diseases in the Caribbean.

## Figures and Tables

**Figure 1 viruses-12-00144-f001:**
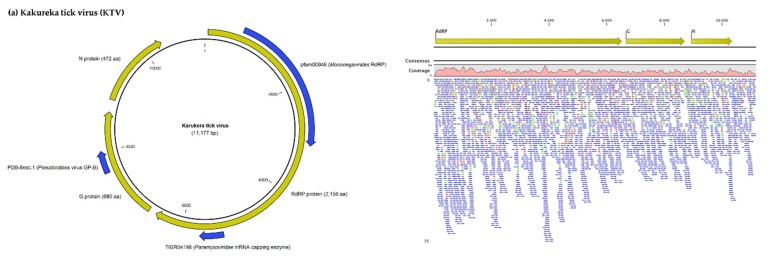
Genomes structure and organization of the two *Chuviridae*, (**a**) Karukera tick virus (**b**) Wuhan tick virus 2 and the *Phenuiviridae* (**c**) Lihan Tick Virus detected in *Rhipicephalus microplus* and *Amblyomma variegatum* ticks collected in Guadeloupe and Martinique. Coding sequences are highlighted with a yellow arrow and pfam functional domains with a blue arrow. Genome horizontal coverage are indicated in pink. For clarity, read mapping for LTV was performed on concatenated segments (represented by a green arrow).

**Figure 2 viruses-12-00144-f002:**
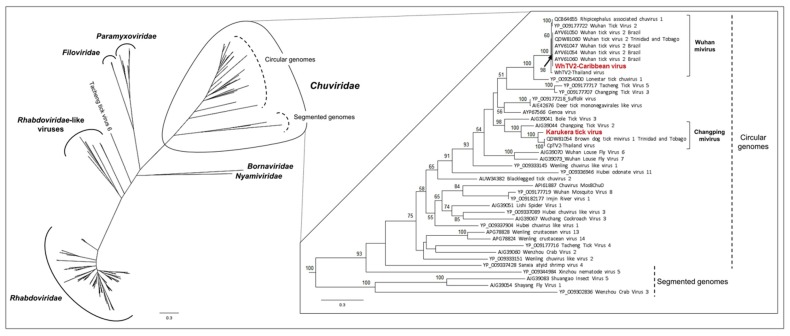
Phylogenetic relationship of *Jingchuvirales*-related viral genomes identified in Caribbean ticks (in red) with other representative viruses from the *Mononegavirales* and *Jingchuvirales* orders (*N* = 169 reference sequences). Nodes with bootstrap values greater than 50 are noted. Phylogenetic reconstruction was performed by Maximum Likelihood on the complete RdRP amino-acid gene (model: LG+G+I+F).

**Figure 3 viruses-12-00144-f003:**
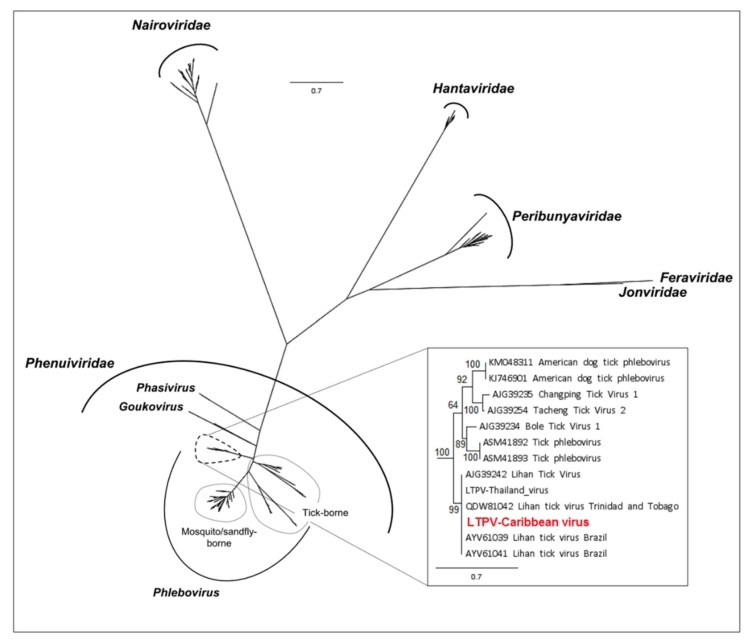
Phylogenetic relationship of *Bunyavirales*-related viral genomes identified in Caribbean ticks (in red) with other representative viruses (*N* = 229 reference sequences). Nodes with bootstrap values greater than 50 are noted. Phylogenetic reconstruction was performed by Maximum Likelihood on the complete RdRP amino-acid gene (model: LG+G+I+F).

**Figure 4 viruses-12-00144-f004:**
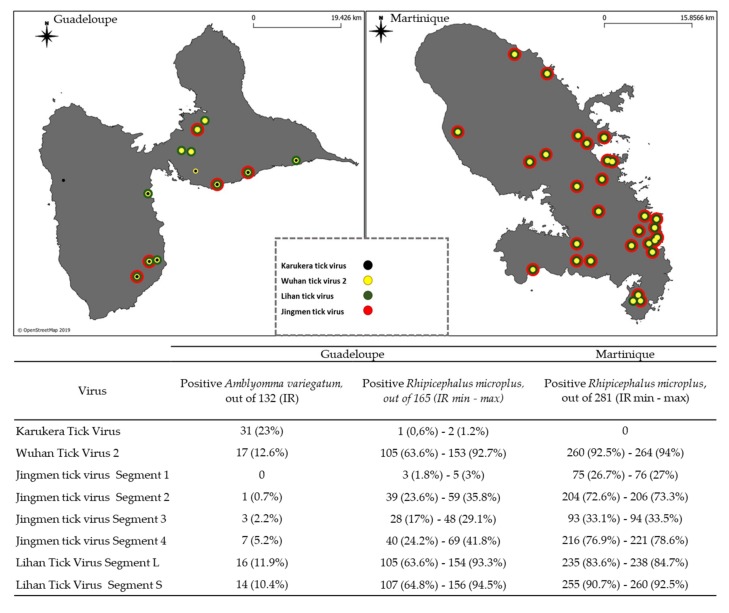
Virus Infection rates in ticks collected in Guadeloupe and Martinique. Number of positive ticks *Amblyomma variegatum* (out of the 132), *Rhipicephalus microplus* from Guadeloupe (out of 165) and Martinique (out of 281). As some samples of *Rhipicephalus microplus* were pooled, we present minimum and maximum infection rates of infected ticks. The maps of Guadeloupe and Martinique represent the ticks’s collection sites found positive for the Karukera tick virus, Wuhan tick virus 2, Jingmen tick virus and Lihan tick virus.

**Figure 5 viruses-12-00144-f005:**
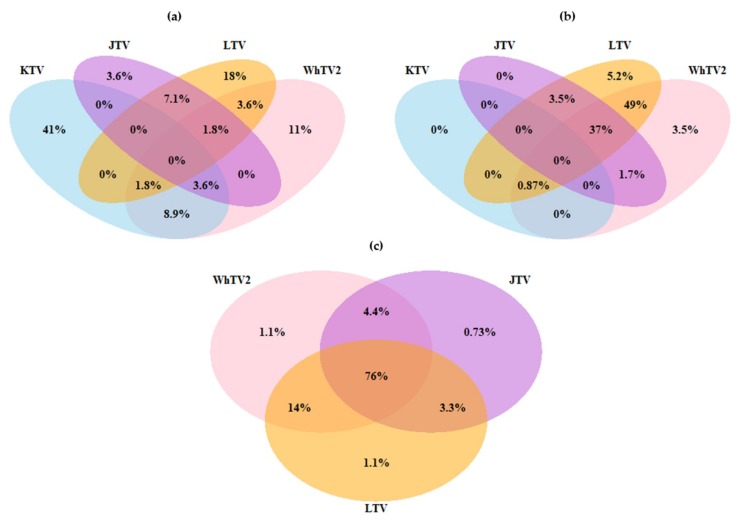
Virus co-infections in Carribean ticks. (**a**) Percentage of viral co-infections in positive *Amblyomma variegatum* samples collected in Guadeloupe (out of 56 positive samples) (**b**) Percentage of viral co-infections in positive *Rhipicephalus microplus* samples collected in Guadeloupe (out of 115 positive samples) and (**c**) Percentage of viral co-infections in positive *Rhipicephalus microplus* samples collected in Martinique (out of 274 positive samples). KTV: Karukera tick virus; WhTV2: Wuhan tick virus 2; JTV: Jingmen tick virus; LTV: Lihan tick virus.

**Table 1 viruses-12-00144-t001:** List of the design developed in this study allowing the detection of the four RNA virus analyzed.

Virus	Target	Design Name	Sequence (5′–3′)	Amplicon Size (bp)
Karukera Tick Virus	Putative RdRP ^1^ gene	KTVL_Poly_F	CACATGTCTCGGAGCGAGG	136
		KTVL_Poly_R	TTCCTGAACGTCTGAGGCTG	
		KTVL_Poly_S	AAAGCTATTCGGGCACGTCATTAAAGTGG	
Wuhan Tick Virus	Putative RdRP ^1^ gene	WTV_Poly_F	GACCCAGGGAGAGTTAGATG	119
		WTV_Poly_R	ACCTGCTGTTCCATGAGCTC	
		WTV_Poly_S	TAGCCCGTAAACTCTTGGGATTTCGTATGC	
Jingmen Tick Virus Segment 1	Putative NS5-like gene	JTV_Seg1_F	ACGTGAAGGAAATATCATTCTGC	100
		JTV_Seg1_R	GCGAATATCTCTCCCACGTC	
		JTV_Seg1_P	TCCCACAGGTACTGGCCGGTAAAGTA	
Jingmen Tick Virus Segment 2	Putative Glycoprotein	JTV_Seg2_F	ATCTTCAGCGCTATCACCGC	95
		JTV_Seg2_R	CGGTTTTGTCGGCGAATGATG	
		JTV_Seg2_P	ATTGCAGCGATGAGTGGGACGAGCG	
Jingmen Tick Virus Segment 3	Putative NS3-like gene	JTV_Seg3_F	CGTGGGGAAGGACAAAAGC	102
		JTV_Seg3_R	CCTTATCTCTCCGCTAGTGG	
		JTV_Seg3_P	AAGGCAGCTTGCATAGAGATGACCGC	
Jingmen Tick Virus Segment 4	Putative membrane protein gene	JTV_Seg4_F	ACAGCGTGCTAGTCTTCGC	79
		JTV_Seg4_R	GGGAGTTGAAAGTGTATGCCA	
		JTV_Seg4_P	AGGCACGTTTGTGATGGTTCAGGACAG	
Lihan Tick Virus Segment L	Putative RdRP ^1^ gene	LTV_SegL_F	ACATGGGTGTATCCAACACAC	127
		LTV_SegL_R	ACCGACATAGCCCATCGAG	
		LTV_SegL_P	ACAGGAGTCTAAACAAGGACGGGTGCAT	
Lihan Tick Virus Segment S	Putative nucleopasid protein (N) gene	LTV_SegS_F	TTGACGTTCTACTCGGCCAC	123
		LTV_SegS_R	TACTGCCTGCGTCATGAGTG	
		LTV_SegS_P	AATTCTAGCCGCTCACCATTCTGCCCA	

^1^ RdRP: RNA-dependent RNA polymerase.

**Table 2 viruses-12-00144-t002:** Main viral sequences identified by NGS in Caribbean ticks. The quantification of abundance of each viral taxon was obtained by summing the length (in nucleotides) of all sequences being associated to this taxon, weighted by the *k*-mer coverage of each contig.

	Family	Genus	Closest Viral Sequence (GenBank Accession Number)	% Identity (aa)	Abundance (nt)
ssRNA+	*Flaviviridae*	unclassified	Jingmen tick virus (MH133317-20)	72%–100%	6,003,829
	*Tymoviridae*	*Marafivirus*	Peach virus D (NC_033828)	85%–86%	1,087,064
			Maize rayado fino virus (NC_002786)	96%–100%	20,750
			Oat blue dwarf virus (NC_001793)	94%	10,556
			Citrus sudden death-associated virus (DQ185573)	93%–100%	3390
			Olive latent virus 3 (NC_013920)	95%–96%	687
		*Maculavirus*	Bee Macula-Like virus 2 (MF998084)	96%	6615
			Grapevine Red Globe virus (KX109927)	92%–100%	1824
			Grapevine fleck virus (NC_003347)	79%–100%	639
		*Tymovirus*	Erysimum latent virus (NC_001977)	95%–100%	2709
		unclassified	Bee Macula-like virus (KT162925)	59%–100%	551,038
			Varroa Tymo-like virus (NC_027619)	87%–100%	502,827
	unclassified *Tymovirales*		Peach virus T (KY348615)	98%	213
			Fusarium graminearum mycotymovirus 1 (KT360947)	100%	72
ssRNA−	*Chuviridae*	*Mivirus*	Wuhan tick virus 2 (NC_028266)	82%–100%	184,236
			Changping tick virus 2 (NC_028260)	58%–95%	1596
		unclassified	Lonestar tick chuvirus 1 (NC_030204)	100%	144
	*Phenuiviridae*	*Phlebovirus*	Lihan tick virus (KM817672 - KM817736)	76%–100%	277,395
unclassified RNA viruses			Hubei sobemo-like virus 15 (NC_032208)	50%–95%	96,054
			Hubei partiti-like virus 7 (KX884117)	80%–83%	147
			Wuhan fly virus 5 (NC_033485)	76%	75
			Wenling chuvirus-like virus 1 (NC_032409)	87%	72
dsRNA	*Partitiviridae*	unclassified	Maize associated partiti-like virus (MF372918)	53%–96%	16,857

**Table 3 viruses-12-00144-t003:** Closest homology for genome viruses and ORF using sequence identity search from the NCBI nucleotide databases with the Basic Local Alignment Search Tool (blastn algorithmes for genome sequences and blastp algorithmes for protein sequences) (October 2019). C%: query coverage (%); I% query identity (%). *: Prototype strain.

Virus	Sequence	Closest Homology	C%	*E*-value	I%	Accession Number
Karukera Tick Virus	Complete genome	Brown dog tick mivirus 1 (Trinidad and Tobago)	89	0	71.7	MN025520.1
	L protein (RNA polymerase)	Polymerase (Mivirus sp.)	100	0	82.4	QDW81054.1
	G protein (Glycoprotein)	Glycoprotein (Mivirus sp.)	99	0	77.1	QDW81055.1
	N protein (Nucleoprotein)	Nucleoprotein (Mivirus sp.)	94	0	61.7	QDW81056.1
Wuhan Tick Virus 2	Complete genome	Wuhan tick virus 2 isolate WTV2_100 (Brazil)	98	0	99.4	MH155927.1
	L protein (RNA polymerase)	Polymerase (Wuhan tick virus 2) *	100	0	98.1	YP_009177722.1
	G protein (Glycoprotein)	Glycoprotein (Wuhan tick virus 2)	100	0	99.1	QDW81058.1
	N protein (Nucleoprotein)	Nucleoprotein (Wuhan tick virus 2)	100	0	99.8	AYV61049.1
Lihan Tick Virus	Complete Segment L	Lihan Tick Virus isolate LTV_L_100 (Brazil)	99	0	99.3	MH155914.1
	L protein (RNA polymerase)	RNA-dependent RNA polymerase (Lihan Tick Virus)	100	0	99.8	AYV61041.1
	Complete Segment S	Lihan Tick Virus strain LH-1 (China) *	100	0	97.8	KM817736.1
	N protein (Nucleoprotein)	Nucleoprotein (Lihan Tick Virus)	99	0	100	AYV61046.1

**Table 4 viruses-12-00144-t004:** Research of endogenous viral elements (EVE) in ticks collected in Guadeloupe and Martinique. Number of positive ticks *Amblyomma variegatum* (out of the 132), *Rhipicephalus microplus* from Guadeloupe (out of 165) and Martinique (out of 281). As some samples of *Rhipicephalus* (B.) *microplus* were pooled, we present minimum and maximum infection rates of infected ticks.

Virus		Guadeloupe	Martinique
	Positive *Amblyomma variegatum,* out of 132 (IR)	Positive *Rhipicephalus microplus*, out of 165 (IR min–max)	Positive *Rhipicephalus microplus*, out of 281 (IR min–max)
Karukera Tick Virus	0	1 (0.6%)–2(1.2%)	0
Wuhan Tick Virus 2	0	41 (24.8%)–59 (35.8%)	87 (31%)–90 (32%)
Jingmenvirus Segment 1	0	0	3 (1.1%)
Jingmenvirus Segment 2	0	0	23 (8.2%)
Jingmenvirus Segment 3	0	4 (2.4%)–7 (4.2%)	8 (2.8%)
Jingmenvirus Segment 4	0	1 (0.6%)–2(1.2%)	14 (5%)
Lihan Tick Virus Segment L	0	1 (0.6%)–2(1.2%)	16 (5.7%)
Lihan Tick Virus Segment S	0	0	1 (0.4%)

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
