# Peer review of "RNA Viruses of Amblyomma variegatum and Rhipicephalus microplus and Cattle Susceptibility in the French Antilles"

_viruses, 2020, doi:10.3390/v12020144_

Round 1

Reviewer 1 Report

In the manuscript “RNA viruses of Amblyomma variegatum and Rhipicephalus microplus and cattle susceptibility in the French Antilles,” the authors describe the results of high-throughput next generation sequencing on the virome of cattle-associated ticks in the French Antilles. The authors identify viral sequences belonging to five different families in the cattle-associated ticks and designed tick screening, and cattle serosurveillance tools to describe the range and prevalence of these viruses. The authors find the individual ticks have a high infection and co-infection rate, however the cattle are seronegative to these viruses suggesting possible restriction to the vector and/or an endosymbiotic relationship. This study is designed and implemented well, the sequencing data is independently verified with RT-PCR and sanger sequencing, and the results fit the conclusions. This information would be of benefit to the scientific community as more HTS studies continue looking into vector viromes. A few minor suggestions for improvement follow.

In lines 88-94 the authors describe the collection of ticks and cattle sera. The dates of collection of ticks occurred in 2014-2015 while the sera was collected mainly in 1994. Do the authors think these discrepant collection dates have any effect on the serosurvey results, or lack of seropositive results?

Line 75, please correct the taxonomy of Estero real virus, it has been reclassified as Nairovirus.

Line 127, please provide NCBI website information for the appropriate database used

Line 132, how many reference sequences were used for multiple alignment for each phylogenetic inference?

Line 140, please describe the “automatic bootstrap replicates” parameter. Convention is to use a specified number of bootstrap replicates.

Author Response

We thanks the reviewer for his useful comments.

In lines 88-94 the authors describe the collection of ticks and cattle sera. The dates of collection of ticks occurred in 2014-2015 while the sera was collected mainly in 1994. Do the authors think these discrepant collection dates have any effect on the serosurvey results, or lack of seropositive results?

The fact that most Guadeloupian cattle sera were collected 20 years before ticks sampling may explain the negative serological result if we suppose that the introduction of tested viruses in the island is recent. Molecular clock determination and calculation of date of introduction may help answer this question, but more representative sequences are needed to accurately determine the date of introduction. In addition, the 22 cattle sera collected in 2019 that also tested negative suggest, if the selected viruses were able to infect cattle, that the seroprevalence is probably low. New collects are therefore needed to answer this question. This observation was added to the discussion, as suggested by the reviewer.

Line 75, please correct the taxonomy of Estero real virus, it has been reclassified as Nairovirus.

Taxonomy corrected accordingly.

Line 127, please provide NCBI website information for the appropriate database used

We used the release of 10/25/2018 for the NCBI/GenBank protein database. This information were added to the Material & Methods section, accordingly.

Line 132, how many reference sequences were used for multiple alignment for each phylogenetic inference?

For the phylogenetic inference of Karukera tick virus and Wuhan tick virus 2 (Figure 2), 169 reference sequences were used. For the phylogenetic inference of Lihan tick virus (Figure 3), 229 reference sequences were selected. These information were added to the legends, accordingly.

Line 140, please describe the “automatic bootstrap replicates” parameter. Convention is to use a specified number of bootstrap replicates.

RAxML (Randomized Axelerated Maximum Likelihood) is a program for ML-based inference of large phylogenetic trees that optimize computing calculation. For example, the “automatic bootstrap replicates” parameter allows the program to automatically halt bootstrapping when some criteria are met, instead of specifying the number of bootstraps for an analysis. Here, the criteria for stopping bootstrap analysis was defined as the difference of likelihood values between replicates less than 0.01%. The analyses were programmed to initially perform 1000 replicates, but the criteria for stopping replicates was achieved before, reducing therefore the duration of analysis. For example for the phylogenetic reconstruction of Jingchuvirales/Mononegavirales (Figure 2), only 204 replicates were necessary; and similarly only 252 replicates were necessary for the phylogenetic inference of Bunyavirales (Figure 3).

Reviewer 2 Report

The manuscript describes a study of the RNA viruses diversity in two tick species, Amblyomma variegatum and Rhipicephalus microplus, collected from cattle susceptibility in Guadeloupe and Martinique. Four viruses detected with high prevalence amongst ticks were further characterized and serologic tests were developed to determine the level of exposure of the cattle. Although no seropositive animals were detected, the complexity of the detected tick virome and the obtained results are important giving new insights in the study of host-microbiome interactions, evolution and zoonotic agents’ transmission.

The major weaknesses and interference factors of the study are clearly discussed, as for instance the fact that most ticks study were engorged.

I only have some specific comments/suggestions regarding:

Table 1 – Jingmen Tick Virus Segment 2, 3 and 4 miss the word virus Table 2 – Giving the complexity in some virus genus assignment and to clarify Best hits, I suggest to add the accession number with the best hit viral sequence, to clearly identify the related viruses. For instance, Bee Macula-like viruses are identified twice and assigned within two different genus, Marafivirus and Tymovirus, and usually Bee Macula-like virus 2 are classified as Maculavirus… Is this a typing mistake or is related to specific sequences of the sub-genomic RNA promoter region (“Tymo-box” or “Marafi-box”) ? Figure 1- The images are too small and it’s impossible to see anything clearly. Images should be enlarged. Figures 6 and 7, are in my opinion not relevant since no additional information is perceived. I would suggest enlarge Figure 1 and remove these two figures. Table S1- for homogenization of presentation, the virus abbreviation should be presented in the correct format for all viral species:

African swine fever virus (ASFV)

Thogoto virus (THOV)

Dhori virus (DHOV)

Kemerovo virus (KEMV

Colorado tick fever virus (CTFV)

Eyach virus (EYAV)

Crimean-Congo Hemorrhagic fever virus (CCHFV)

Dugbe virus (DUGV) … and so forth.

Author Response

We thanks the reviewer for his useful comments.

Table 1 – Jingmen Tick Virus Segment 2, 3 and 4 miss the word virus.

We added the word “virus” accordingly.

Table 2 – Giving the complexity in some virus genus assignment and to clarify Best hits, I suggest to add the accession number with the best hit viral sequence, to clearly identify the related viruses. For instance, Bee Macula-like viruses are identified twice and assigned within two different genus, Marafivirus and Tymovirus, and usually Bee Macula-like virus 2 are classified as Maculavirus… Is this a typing mistake or is related to specific sequences of the subgenomic RNA promoter region (“Tymo-box” or “Marafi-box”)?

As requested, we modified Table 2 to add the accession number of the closest viral sequence for each viruses identified. To answer to the reviewer, the assignation of Bee Macula-like virus in two different genera was a problem when formatting the table: Bee Macula-like virus (reference genome KT162925) is taxonomically identified as unclassified Tymoviridae, and is distinct from Bee Macula-like virus 2 (MF998084) which is classified as a Maculavirus member.

Figure 1- The images are too small and it’s impossible to see anything clearly. Images should be enlarged. Figures 6 and 7, are in my opinion not relevant since no additional information is perceived. I would suggest enlarge Figure 1 and remove these two figures.

As suggested, we enlarged Figure 1 and moved Figure 6 and 7 to supplementary data as Figure S1 and S2 respectively.

Table S1- for homogenization of presentation, the virus abbreviation should be presented in the correct format for all viral species.

As suggested, we added virus abbreviation to each viral species.

Reviewer 3 Report

This paper describes the metagenomic analysis of ticks collected from Caribbean cattle.  This commendable task utilized state of the art technology.  The biggest concern; however, was the use of partially engorged ticks.  How can the authors separate exposure (i.e. viruses in the blood meal) from infection (i.e. virus replicating in the ticks.  It should be clear in the text that the what is being detected is not necessarily a demonstration of a virus-vector-host relationship.

Ln 288-290:  The paper cited here does not agree with this statement.  There appear to be variability of relative transcription rates among bunavirales RNA Segments.

Author Response

We thanks the reviewer for his useful comments.

The biggest concern however, was the use of partially engorged ticks. How can the authors separate exposure (i.e. viruses in the blood meal) from infection (i.e. virus replicating in the ticks. It should be clear in the text that the what is being detected is not necessarily a demonstration of a virus-vector-host relationship.

We added this statement in the discussion accordingly, line 415-416.

Line 288-290: The paper cited here does not agree with this statement. There appear to be variability of relative transcription rates among bunavirales RNA Segments.

We corrected the statements accordingly, using only the cited paper as an example in the literature of a variability of relative transcription rates among viral RNA Segments, Line 286-291